# Role of Neurotrophins in Orofacial Pain Modulation: A Review of the Latest Discoveries

**DOI:** 10.3390/ijms241512438

**Published:** 2023-08-04

**Authors:** Francesca Bonomini, Gaia Favero, Stefania Castrezzati, Elisa Borsani

**Affiliations:** 1Division of Anatomy and Physiopathology, Department of Clinical and Experimental Sciences, University of Brescia, 25123 Brescia, Italy; francesca.bonomini@unibs.it (F.B.); gaia.favero@unibs.it (G.F.); stefania.castrezzati@unibs.it (S.C.); 2Interdepartmental University Center of Research “Adaptation and Regeneration of Tissues and Organs (ARTO)”, University of Brescia, 25123 Brescia, Italy; 3Italian Society of Orofacial Pain (Società Italiana Studio Dolore Orofacciale—SISDO), 25123 Brescia, Italy

**Keywords:** orofacial pain, neurotrophins, NGF, BDNF, trigeminal ganglion, trigeminal nucleus

## Abstract

Orofacial pain represents a multidisciplinary biomedical challenge involving basic and clinical research for which no satisfactory solution has been found. In this regard, trigeminal pain is described as one of the worst pains perceived, leaving the patient with no hope for the future. The aim of this review is to evaluate the latest discoveries on the involvement of neurotrophins in orofacial nociception, describing their role and expression in peripheral tissues, trigeminal ganglion, and trigeminal nucleus considering their double nature as “supporters” of the nervous system and as “promoters” of nociceptive transmission. In order to scan recent literature (last ten years), three independent researchers referred to databases PubMed, Embase, Google Scholar, Scopus, and Web of Science to find original research articles and clinical trials. The researchers selected 33 papers: 29 original research articles and 4 clinical trials. The results obtained by the screening of the selected articles show an interesting trend, in which the precise modulation of neurotrophin signaling could switch neurotrophins from being a “promoter” of pain to their beneficial neurotrophic role of supporting the nerves in their recovery, especially when a structural alteration is present, as in neuropathic pain. In conclusion, neurotrophins could be interesting targets for orofacial pain modulation but more studies are necessary to clarify their role for future application in clinical practice.

## 1. Introduction

Orofacial pain is associated with head, neck, and oral structures. Its assessment in humans is complex, with important knowledge gaps in this field. The neural mechanisms are not yet completely understood; moreover, the available treatments are not fully effective and have limited clinical use due to their adverse effects [1]. From an anatomical point of view, the trigeminal system represents the key structure in orofacial nociceptive transmission. In this regard, its neurochemistry is finely regulated by numerous factors, among which are neurotrophins. They act on the peripheral nerve endings present in the tissues of the orofacial region and influence the neurotransmission in two fundamental anatomical structures: the trigeminal (or Gasserian) ganglion (TG) of the peripheral nervous system and in the trigeminal nucleus (TN) of the brainstem. From an anatomical, morphological, and neurochemical point of view, Prof. L.F. Rodella described the role of these anatomical key points in different studies and books [2,3,4,5,6,7]. Most of the research in this field utilized animal models involving dorsal root ganglia (DRGs) and the spinal cord, leaving the orofacial region as a secondary investigation probably due to the lack of animal models and the more complex anatomy with respect to the spinal one. 

Furthermore, strong evidence indicates a neurological impact of COVID-19, which has pain as a significant symptom, in both the acute phase and the later stages of the disease. Multiple potential mechanisms through which COVID-19 could induce changes in nociceptor excitability have been hypothesized [8], further supported by an altered expression of neurotrophins in serum and saliva from COVID-19 patients [9,10]. In particular, the COVID-19 pandemic has been documented to have influenced orofacial pain perception due to neurological modulation and psychological distress with a consequent impact on the ongoing research results during that specific period [11,12]. The role of neurotrophins in this particular context is to date lacking in the literature.

The aim of this review is to evaluate the involvement of neurotrophins in orofacial nociception with the objective of identifying the key research findings of the last ten years. A summary of the latest discoveries on this topic, to our knowledge, is lacking in the literature, and a comprehensive view could lead and inspire future research directions.

### 1.1. Orofacial Pain Classification

The International Association for the Study of Pain (IASP) defines pain as an unpleasant sensory and emotional experience associated with actual or potential tissue damage or described in terms of such damage [13]. Nociceptive pain, neuropathic pain, and migraine are three common forms of pain disorders, and all pain syndromes could perhaps be included in one of these three categories [14]. Nociceptive pain is defined as pain induced in an intact nervous system by a noxious stimulus originating outside the nervous system. Neuropathic pain is defined as pain initiated or caused by a primary lesion or dysfunction in the nervous system [15]. Migraine is a common head pain syndrome with a neural origin. In particular, a specific and detailed classification of orofacial pain has been recently updated [16].

### 1.2. Orofacial Pain Differential Diagnosis

Recently, a detailed classification of orofacial pain has been published, titled “International Classification of Orofacial Pain, 1st edition (ICOP)” [16]. This document represents a fundamental point of reference for researchers and professionals involved in orofacial pain management. ICOP is particularly useful when the diagnosis is uncertain, or when the clinician is unaware that such a clinical presentation exists. All detailed information presented in the document is organized utilizing the criteria for “diagnosis” of (1) orofacial pain attributed to disorders of dentoalveolar and anatomically related structures, (2) myofascial orofacial pain, (3) temporomandibular joint (TMJ) pain, (4) orofacial pain attributed to lesion or disease of the cranial nerves, (5) orofacial pains resembling presentations of primary headaches, (6) idiopathic orofacial pain, (7) psychosocial assessment of patients with orofacial pain. In particular, in point 4, the pain attributed to a lesion or disease of the trigeminal nerve has been subsequently classified as trigeminal neuralgia (classical), secondary trigeminal neuralgia, and idiopathic trigeminal neuralgia.

### 1.3. Anatomical Structures Involved in Orofacial Nociception

Regarding the anatomical structures involved, the TG is a cluster of primary sensory neurons classified according to their diameter, of which the small ones are primarily involved in nociception. TG neurons carry sensory information from the face and head to the TN. The TN is part of the entire brainstem merging with the cervical spinal cord, and projects sensory information to the thalamus either directly via the trigeminal-thalamic pathway [17] or indirectly via polysynaptic pathways that may involve reticular formation [18,19,20]. The TN is divided into three parts, rostral to caudal: the mesencephalic nucleus (Me5), the primary nucleus (Pr5), and the spinal TN (Sp5). Sp5 is the largest, with the caudal end merging with the cervical spinal cord and divided into three subnuclei [21]: oralis (Sp5O), interpolaris (Sp5I), and caudalis (Sp5C) [16,18,19]. Among them, Sp5C is the major relay for trigeminal nociception in which neurons receive a consistent innervation from unmyelinated and thinly myelinated peptidergic sensory fibers arising from all three divisions of the trigeminal nerve [22,23,24]. Sp5O and Sp5I receive only a sparse innervation by nociceptive fibers [25,26,27,28] (Figure 1). In addition, it has also been reported that the nociceptive inputs from receptors in deep craniofacial tissues are relayed to the ventral Sp5I/Sp5C transition region through the Sp5C/cervical dorsal horn C2 junction region [29]. Recent studies demonstrate that orofacial injury and noxious stimulation of dental and craniofacial regions activate a distinct region in the Sp5I/Sp5C transition zones [30,31,32].

### 1.4. Neurochemistry of Orofacial Nociception: The Neurotrophins

Considering the neurochemistry, neurotrophins are able to modify the strength of neuronal connections through the precise modulation of presynaptic activity, directly enhancing quantal neurotransmitter release [33] through two different classes of receptors: the receptor family of tyrosine kinases (Trk) and the p75 neurotrophin receptor (p75NTR), a member of the tumor necrosis factor (TNF) receptor superfamily [34] (Figure 2). The neurotrophin family includes nerve growth factor (NGF), brain-derived neurotrophic factor (BDNF), neurotrophin-3 (NT-3), and neurotrophin-4 (NT-4) [35], also known as neurotrophin-5 (NT-4/5) [35,36]. Specifically, NGF binds with a high affinity to tropomyosin receptor kinase A (TrkA); BDNF and NT-4 bind with a high affinity to tropomyosin receptor kinase B (TrkB), and NT-3 binds with a high affinity to tropomyosin receptor kinase C (TrkC) [37]. Apart from the activation of TrkC, NT-3 also activates TrkA and TrkB, albeit with lower affinities [38], and all mature neurotrophins bind to the p75NTR with similar affinity [36]. The expression patterns of Trk receptors in sensory neurons of mouse TG have been characterized during neurogenesis [39,40]. Although the expression of Trk receptors remains unchanged in most sensory neurons, a small percentage of neurons show dynamic shifts of neurotrophin receptors and neurotrophin dependence [41].

Neurotrophins are involved in regulating the survival, neuroplasticity, and maintenance of phenotype, growth, and development of neurons. Among them, NGF is considered essential for these functions [14,42] but also with a well-known role as a mediator of pain, itch, and inflammation [43,44,45] as well as a minor role in antinociception [46]. In addition, BDNF has a role in nociceptive excitability modulation [47,48] and a minor role in antinociception [49,50,51,52]. NT-3 has primarily an antinociceptive role [14] and a minor role in pronociception [53,54], while NT-4 has a strictly proprioceptive role [14]. Also, NT-6 and NT-7 have been described, but the genes have been identified only in fish and probably do not have any mammalian orthologues [55,56]. In cultured Xenopus neuromuscular junctions, BDNF and NT-3 were initially reported to promptly enhance synaptic transmission at the presynaptic level [57]. It was subsequently demonstrated that BDNF increases the frequency of spontaneous synaptic transmission [58]. NGF-TrkA signaling is mostly associated with the upregulation of transient receptor potential vanilloid 1 (TRPV1) [59], substance-P (SP), calcitonin gene-related peptide (CGRP), and the sodium channels, Nav1.8 and Nav1.9 [14]. BDNF-TrkB signaling influences several downstream signaling cascades (e.g., PLC-PKC, PI3K-Akt, Raf-Erk) and promotes *N*-methyl-d-aspartate (NMDA) receptor activation and downregulation of the potassium chloride cotransporter 2, which is the potassium-chloride exporter in neurons [60] and/or GABAergic inhibitory signaling mechanisms [61]. In neuropathic pain, the p75NTR downstream signaling activation seems to involve the c-Jun-NF-kB pathway and may potentiate Trk-mediated signaling [14]. Neurotrophin receptors’ downstream signaling is summarized in Figure 3.

The source of neurotrophins is heterogeneous, from the nervous system target organs to the nervous system itself. NGF is synthesized and secreted by sympathetic and sensory target organs [62]. Subsequently, it is captured in nerve terminals by receptor-mediated endocytosis, and it is transported through axons to neuronal cell bodies where it acts to promote neuronal survival and differentiation. Moreover, many neurons also synthesize neurotrophins. For example, several populations of sensory neurons have been shown to synthesize BDNF [63,64]. All four neurotrophins and their receptors are synthesized in the central nervous system, although at different levels and with different regional distributions [65,66,67]. In the brain, the neurons are considered the major cellular source of neurotrophins, in particular BDNF [68,69,70]. Moreover, astrocytes [71] and microglial cells have been found to be another physiological source of neurotrophins [65,72,73]. Some evidence addressed the hypothesis that BDNF may act in an autocrine or paracrine fashion to support DRG sensory neurons [74,75]. In other instances, it may be transported anterogradely and act trans-synaptically on the central afferents of these neurons within the brain [64]. In addition, infiltration of macrophages during an inflammatory state, due to peripheral nerve injury, promotes cytokine release, which in turn, induces NGF synthesis in Schwann cells and fibroblasts. NGF and other neurotrophic factors synthesized in damaged nerves are believed to be essential for the survival and regeneration of injured neurons. Finally, when overexpressed in the skin, sufficient target-derived NGF is released from the soma of trigeminal sensory neurons to support aberrant innervation by NGF-dependent sympathetic fibers [76]. Thus, in some circumstances, neurotrophins provided by one cell are not only effective in supporting neurons, whose axons are in its vicinity, but they also can provide support to more distant neurons via transcellular transport.

## 2. Research Findings on Neurotrophins during the Last 10 Years (2013–2023)

This review investigates the role of neurotrophins in orofacial nociceptive transmission and their underlying cellular mechanisms. The articles have been selected utilizing the following criteria (Figure 4). Three independent researchers scanned the articles in the following databases: PubMed, Embase, Google Scholar, Scopus, and Web of Science. The keywords used for the query were “neurotrophins” (“NGF”, BDNF”, “NT-3”, “NT-4”), “orofacial”, “pain” (“pain”, “nociception”, “hyperalgesia” and “sensitization”) and “trigeminal ganglia” or “trigeminal nucleus”. This search of articles published during the last 10 years (from 2013 to 2023) elicited 90 papers. Later, the papers were screened for eligibility according to the inclusion and exclusion criteria. Exclusion criteria were review articles or clinical trials not registered or published. Inclusion criteria were original research articles that were registered and published clinical trials with a randomized controlled trial design or a non-randomized controlled trial design. Finally, 56 articles were excluded, and 33 articles were selected. Among the selected articles, 29 were original research articles and 4 were clinical trials. The included studies were analyzed, summarized, and discussed according to the purpose of the present review. 

Finally, the selected manuscripts were subdivided considering the anatomical topic (peripheral orofacial tissues, TG, and brainstem) and the kind of pain (nociceptive, neuropathic, and migraine). Seven articles overlapped because more than one anatomical area was considered.

### 2.1. Neurotrophins in Peripheral Orofacial Tissues

The neurotrophins’ role in orofacial pain modulation starts in the peripheral tissues, for example, cheek skin, muscle, periodontium, gingiva, and teeth. The data reported were obtained from 9 original research articles and 3 clinical trials.

#### 2.1.1. Nociceptive Orofacial Pain Modulation

In this section, 7 original research articles (on animal models and human samples) and 3 clinical trials were included and analyzed. 

The first work investigated rat TG neurons projecting to the gingivomucosa and dental pulp [77]. The results of this original research article showed a different involvement of the small-sized neurons innervating the gingivomucosa (14%) with respect to dental pulp (5%). Moreover, TrkA-positive neurons were, respectively, 76% and 86% of the total gingival or pulpal neurons, underlying that these peripheral structures were richly innervated by nociceptive TrkA-expressing neurons. In addition, almost all of the pulpal neurons were larger NGF-dependent A-fiber nociceptors without affinity to bind isolectin B_4_ (IB_4_), a marker for non-peptidergic small neurons, while almost half of the gingival neurons were smaller IB_4_-binding C-fiber nociceptors. So, the different sensitivity of both tissues during normal and pathological conditions could be explained in part by the difference in the phenotype of sensory neurons [77].

The injection of NGF (25 µg/mL, 10 µL) into the masseter muscle induced a reduction in the mechanical withdrawal threshold for 1 day in male rats and 5 days in female rats [78]. These results were in accordance with the data in humans where a sex-related difference has been found in NGF-induced mechanical sensitization. Indeed, intramuscular injection with the competitive NMDA receptor antagonist DL-2-amino-5-phosphonovaleric acid (0.020 g/mL, 10 µL) reversed the mechanical sensitization in male but not in female rats [78]. 

Furthermore, considering that women with reproductive capability are more likely to suffer from temporomandibular disorders (TMD), estradiol’s influence on synoviocyte gene expressions involved in the allodynia of the inflamed temporomandibular joint (TMJ) in rats was explored [79]. The influence of 17-β-estradiol on NGF and TRPV1 expression in TMJ synovium was determined in vivo and in vitro and analyzed by Western blot and real-time PCR. TMJ arthritis was induced by an injection of Complete Freund’s adjuvant into a rat, and the head withdrawal threshold was examined using a von Frey Aesthesiometer. The results showed that the expressions of TRPV1 and NGF were upregulated by 17-β-estradiol in a dose-dependent manner [79]; moreover, intra-TMJ injection of TRPV1 antagonist, capsazepine, significantly attenuated the allodynia of the inflamed TMJ. In addition, the in vitro evaluation of primary synoviocytes showed the TRPV1 upregulation by NGF, lipopolysaccharide, and 17-β-estradiol, while NGF antibodies fully blocked lipopolysaccharide and 17-β-estradiol-induced upregulation of TRPV1. This article presented a possible local mechanism for the involvement of 17-β-estradiol in TMJ inflammation, with the potential for use in clinical practice helping to understand the sexual dimorphism of TMD pain [79].

Another double-blind, placebo-controlled randomized clinical trial with capsaicin nasal spray was conducted in patients with idiopathic rhinitis (*n* = 33) and healthy control subjects (*n* = 12), exploring the therapeutic action of capsaicin treatment, which was based on the ablation of TRPV1–SP nociceptive signaling pathway [80]. Patients were also evaluated considering the mRNA expression of NGF in nasal biopsy specimens. The data showed that NGF mRNA expression was unchanged while TRPV1 was higher compared to the healthy control subjects. In conclusion, these results did not link NGF to idiopathic rhinitis [80].

Additionally, a double-blind, randomized placebo-controlled clinical trial recruited 16 healthy participants who were injected for 2 days with NGF into the masseter and temporalis muscles and isotonic saline on the contralateral side [81]. Practicing repeated intramuscular injections of NGF, an increase in mechanical sensitivity for the masseter muscle but not the temporalis muscle was observed; nevertheless, both referred pain frequency and the number of headache days did not increase following NGF injections [81]. These findings support the hypothesis that mechanical sensitization in the masseter and temporalis muscles differs following injections of NGF, but the referred pain and headache frequency did not seem to be related to NGF sensitization in this model. In conclusion, the reported results supported the idea that, in healthy individuals, referred pain may be an epiphenomenon of the muscle in response to noxious input [81].

The concentrations of NGF and BDNF, together with glutamate, serotonin, and SP, in saliva and plasma from a well-defined group of patients with chronic TMD-myalgia (*n* = 39) and in a group of pain-free controls (*n* = 39) in a clinical examination have been evaluated [82]. Salivary NGF and BDNF were lower in patients compared to controls. Only plasma BDNF was higher in patients than in controls. The obtained results need further investigation to better explore their correlations to psychological distress [82].

Moreover, in humans, the effect of NGF injection on the density and expression of NGF, SP, and NMDA-receptors by the nerve fibers in the masseter muscle was explored to correlate expression with pain characteristics and to determine any possible sex-related differences in these NGF effects [83]. The study involved healthy females (*n* = 15) and age-matched healthy males (*n* = 15) (mean ± SD age: 30 ± 12 years). Women displayed a greater magnitude of NGF-induced mechanical sensitization that was also associated with nerve fibers’ expression of NMDA-receptors when compared to men. So, these findings suggested that increased peripheral NMDA-receptor expression could be associated with masseter muscle pain sensitivity in women [83].

Interestingly, the role of NGF in tooth mechanical hyperalgesia in a rat model was investigated to elucidate the underlying mechanisms [84]. Tooth mechanical hyperalgesia was induced by ligating closed coil springs between incisors and molars in Sprague Dawley rats. Retrograde labeling was performed by the periodontal administration of fluor-conjugated NGF and the detection of fluorescence in TG. The results revealed that tooth movement elicited tooth mechanical hyperalgesia that could be alleviated by NGF-neutralizing antibodies. Furthermore, NGF was upregulated in periodontium (mainly in periodontal fibroblasts) and TG [84]. Retrograde labeling revealed that periodontal NGF was retrogradely transported to TG after day 1. Acid-sensing ion channel 3 (ASIC3; mechanical sensory channel) and NGF were co-expressed in trigeminal neurons and the percentage of co-expression was significantly higher following tooth movement. Taken together, the data showed that, in response to force stimuli, periodontal fibroblasts upregulated the expressions of NGF, which was retrogradely transported to TG, where NGF elicited tooth mechanical hyperalgesia through upregulating ASIC3. In conclusion, NGF-based gene therapy was proposed as a viable method for alleviating tooth-movement-induced mechanical hyperalgesia [84].

A randomized, double-blinded placebo-controlled study investigated the underlying pathway of bruxism that could represent an overlearned behavior due to the absence of corticomotor plasticity following a relevant tooth-clenching task [85]. An experimental pain/sensitization model in humans caused by intramuscular NGF administration in combination with a standardized tooth-clenching task was used. Participants characterized as definitive bruxers (*n* = 15) or controls (*n* = 13) were randomly assigned to have either NGF or isotonic saline injected into the right masseter muscle. This study was the first which showed that bruxers did not significantly change the central modulation of motor pathways as a consequence of NGF-induced sensitization in combination with a motor training task while in control participants substantial changes in corticomotor excitability were observed. These preliminary findings could have therapeutic implications to manage bruxism, nevertheless, further studies on larger sample sizes will be needed [85].

In addition, the tumor-released BDNF contribution to oral cancer pain via peripheral TrkB activation was studied in a mouse orthotopic xenograft model (tongue tumor) [86]. It is important to consider that the quality of life of head and neck cancer patients is notoriously poor due to the greater prevalence of pain compared to other cancer types. It was shown that locally blocking BDNF reversed tongue tumor-induced pain behaviors in vivo and that peripheral TrkB receptors had a key role in this condition [86].

#### 2.1.2. Neuropathic Orofacial Pain and Migraine Modulation

In this section, 2 original research articles on animal models and 0 clinical trials were included and analyzed. 

The oldest study investigated the NGF’s ability to influence mature nociceptors phenotype by altering the synthesis of neuropeptides in an orofacial neuropathic animal model (chronic constriction injury of the rat mental nerve) [87]. The nerve fiber skin density analysis (from day 11 to 21) underlined a temporal mismatch in behavior, skin NGF, and phenotypic changes in sensory nerve fibers. The data indicated that an increase in NGF did not cause hyperalgesia; however, it could contribute to the altered neurochemistry of cutaneous nerve fibers [87].

More recently, the increased expression of NGF and S100B, neurotrophic factors fundamental for nerve cells survival, has been reported following the injection of Freeze-Dried Platelet Rich Plasma (FD-PRP) in a rat model of infraorbital axonotmesis injury [88]. The treated group showed a significant increase in both NGF and S100B at the infraorbital nerve level compared to the controls, ultimately, demonstrating the effect of FD-PRP in inducing neuroregeneration [88].

### 2.2. Neurotrophins in Trigeminal Ganglia

The data reported was obtained from 17 original research articles and 1 clinical trial.

#### 2.2.1. Nociceptive Orofacial Pain Modulation

In this section, 11 original research articles on animal models and 0 clinical trials were included and analyzed. The studies are presented following temporal criteria (from the oldest to the newest).

The studies, previously described in the peripheral tissues section assessing TG neurons of rats projecting to the gingivomucosa and dental pulp [77] and the alterations in TG after the injection of NGF into the masseter muscle [78] are also considered in this section. In particular, NGF was able to increase the number of NMDA receptor subunit subtype 2B (NR2B)-expressing rat trigeminal masseter ganglion neurons in both sexes, but the increased expression of neuropeptides (CGRP and SP) was observed in NR2B-expressing masseter ganglion neurons only in female rats [78]. In healthy animals, similar basal expression levels of NR2B and SP were found in peripheral fibers from the masseter muscle of both males and females [78].

In addition, the functional significance of hyperalgesia related to the BDNF-TrkB signaling system in TG neurons projecting to the Sp5I/Sp5C transition zone following masseter muscle inflammation has been investigated [48]. The average number of BDNF/TrkB-immunoreactive small/medium-diameter TG neurons was significantly higher in inflamed rats than in naïve rats. In whole-cell current-clamp experiments, the majority of dissociated small-diameter TG neurons showed a depolarization response to BDNF that was associated with spike discharge, and the concentration of BDNF that evoked a depolarizing response was significantly lower in the inflamed rats [48]. In addition, the relative number of BDNF-induced spikes during the current injection was significantly higher in inflamed rats. The BDNF-induced changes in TG neuron excitability were abolished by Trk inhibitor, K252a. BDNF enhanced the excitability of the small-diameter TG neurons projecting onto the Sp5I/Sp5C following masseter muscle inflammation. These findings suggested that ganglionic BDNF-TrkB signaling could be a therapeutic target for the treatment of trigeminal inflammatory hyperalgesia [48].

Furthermore, the developmental acquisition of transient receptor potential ankyrin 1 (TRPA1) in the cornea of an animal model (white Leghorn’s eggs) was examined [89]. Corneal nerves protect the cornea and preserve vision by transducing damaging stimuli as sensations of pain. TRPA1s are membrane proteins that form a channel responsible for nociception. TRPA1 mRNA showed a progressive temporal increase in the ophthalmic lobe of the TG in vivo during embryonic development [89]. Among the results in vitro, the addition of exogenous NT-3 protein to cultured ganglia greatly increased the expression of TRPA1. NT-3 is involved in the increased TRPA1 expression in embryonic day 8 ganglia. One conclusion of this study was that NT-3 had an important role during the process of nociceptor specification and described for the first time a relationship between NT-3 and TRPA1 regulation [89].

In addition, an RNA sequencing assay in rat TG was performed to identify transcriptome profiles of genes relevant to hyperalgesia following inflammation of the rat masseter muscle [90]. Masseter inflammation differentially regulated more than 3500 genes in TG. Inspection of individual gene expression confirmed the transcriptional changes of multiple nociceptor genes associated with masseter hyperalgesia such as *Bdnf*, *Trpv1*, *Trpa1*, *P2rx3* (purinergic receptor P2X3), and also suggested various novel probable contributors [90].

Moreover, the nociceptive role of BDNF in TG was supported by a study in transgenic mice for the expression of the downstream regulatory element antagonist modulator (DREAM) protein, a multifunctional calcium-binding protein that acts in the nucleus as a calcium-dependent transcriptional repressor [17,91].

Interestingly, p75NTR targeted lentiviral interference therapy was proposed to alleviate trigeminal nociception to achieve targeted depletion of TRPV1 in rat TG sensory neurons with novel and positive results monitoring TG and Sp5C [92]. This study introduced a therapeutic targeting strategy that provided a means to post-transcriptionally downregulate TRPV1 in TG nociceptors of adult rats in vivo [92].

Furthermore, it has been reported a gene upregulation of *Bdnf* in mice TG as a consequence of an orthodontic force producing mechanical irritation and localized inflammation in the periodontium, a cause of pain in most patients [93].

Recently, in an in vivo study in rats, the NGF involvement in tooth mechanical hyperalgesia was explored to elucidate the underlying mechanisms previously described in the peripheral tissues section [84]. In particular, the authors indicated that NGF knockdown at the TG level promotes the downregulation of ASIC3, which in turn is able to modulate orofacial pain induced by tooth movement in both periodontal tissues and TG. Notably, ASIC3 and NGF were co-expressed in trigeminal neurons and this co-expression was increased after tooth movement. In conclusion, gene therapy based on NGF could be a promising “tool” against tooth mechanical hyperalgesia (Figure 5) [84]. 

In addition, using a known and documented in vivo model of orofacial pain induced by orthodontic tooth movement, the role of V-type proton ATPase subunit isoform 1 (Atp6v0a1) and NGF-CGRP was evaluated [94]. Orthodontic force produces mechanical irritation and localized inflammation in the periodontium. In detail, orofacial pain was elicited by ligating closed-coil springs between incisors and molars in Sprague Dawley rats. Therefore, animals were subjected to NGF or anti-NGF antibody solutions (both at a final concentration of 0.1 µg/µL) in TG. Lentivirus vectors carrying Atp6v0a1 shRNA were used to knock down the expression of Atp6v0a1 in the TG and in SH-SY5Y neurons. This study revealed that NGF intraganglionic administration increased the nociceptive CGRP level in TG and Sp5C, exacerbating orofacial pain. However, neutralizing anti-NGF antibodies significantly reduced orofacial pain [94]. Notably, Atp6v0a1 shRNA was able to reduce the CGRP level, alleviating orofacial pain related to orthodontic tooth movement. The authors consequently proposed a possible orofacial pain molecular pathogenetic mechanism characterized by the modulation of orofacial pain through the NGF neurotrophin that, in turn, regulates CGRP expression both in TG and Sp5C. Furthermore, the authors revealed that Atp6v0a1 was involved in the nociceptive transmission related to orofacial pain (Figure 5) [94].

Finally, the results on TG-isolated neurons indicated that BDNF enhances T-type currents through the stimulation of TrkB coupled to PI3K-p38-PKA signaling, consequently inducing neuronal hyperexcitability and pain hypersensitivity in rats [95].

As reported in the previously described original studies, the regulatory mechanisms of inflammatory orofacial pain are multiple and complex, and further studies are needed to better clarify these mechanisms and find possible key factors to counteract pain.

#### 2.2.2. Neuropathic Orofacial Pain and Migraine Modulation

In this section, 7 original research articles in animal models and 1 clinical trial were included and analyzed. The studies are presented following temporal criteria (from the oldest to the newest).

A very common complaint observed in the orthodontic field is orofacial pain triggered by orthodontic forces related to periodontal inflammatory responses. Therefore, the TG is stimulated to produce various inflammatory factors, which, in turn, altered the TG neurons’ activity promoting pain sensitivity [96]. Among these factors, the chemokine (C-C motif) ligand 19 (CCL19) is a known mediator of synovial inflammation and is involved in orofacial neuropathic pain [96]. CCL19 is a small cytokine belonging to the CCL chemokine family, and it is engaged in various inflammatory disorders. Furthermore, CCL19 is a known chemokine involved in pain regulation and neuroinflammatory responses [96]. Recently, a study on NGF involvement in neuropathic orofacial pain modulation was performed using a tooth movement orofacial pain animal model, indicating neuroinflammation [96]. Sprague Dawley rats were subjected to closed-coil springs fixed in place with ligation wires between the left maxillary first molar and the upper incisor with the aim to stimulate tooth movement. The authors reported that mRNA and protein CCL19 levels in the presence of NGF were elevated, whereas the expression of CCL19 mRNA and protein in rats with anti-NGF inoculation was significantly decreased. Moreover, rats injected with NGF plus anti-CCL19 antibodies presented significantly higher bite force than those treated only with NGF, underlining that CCL19 may interact with NGF and consequently contribute to attenuate tooth-movement-related pain. The authors concluded that there is a strict link between CCL19 chemokine and orofacial pain; in addition, NGF can modulate tooth-movement-induced mechanical hyperalgesia [96]. Further studies are needed to determine the relationship between the expression of this chemokine with the postoperative prognosis of orthodontic treatment as well as whether NGF regulates not only CCL19 but also other related molecular pathways correlated to tooth-movement pain [96]. 

Moreover, the effects of BDNF on the neuronal excitability of TG neurons and the pain sensitivity of rats mediated by T-type calcium channels were investigated [81]. 

Interestingly, no association was found between TrkA gene polymorphisms and trigeminal neuralgia in humans [97].

Furthermore, in another original research article, how weak electrical stimulation modulates the neuroprotective/neuroregenerative and functional processes of rat TG neurons was investigated by studying the expression of neurotrophins and Glia-Derived Neurotrophic Factors (GDNF) receptors [98]. RET, proto-oncogene tyrosine kinase “rearranged during transfection”, is expressed in all ganglia (from 25% in human TG to 60% in mouse DRG); moreover, the RET-positive cell population is characterized by mechanoreceptive neurons [98,99]. Neurostimulation was applied to the proximal stump of a transected left infraorbital nerve. Regarding neurotrophins, the results showed the involvement of RET-expressing small and large neurons, which include thermo-nociceptors and mechanoreceptors, as well as on the IB_4_- and TrkB-positive populations. 

In addition, the expression of BDNF and GDNF was investigated in the TG and in the brainstem transitional zone of trigeminal primary afferent pathways in a trigeminal neuropathic rat model induced by mechanical compression [100]. Regarding BDNF, it increased in TG and decreased in the brainstem in neuropathic animals, suggesting BDNF involvement in nociceptive transmission [100]; the data on TG were furtherly supported by another study in rats [101].

Another interesting study regarded the regulatory effect of NMDA receptors in mediating peripheral and central sensitization in a mice model of orofacial neuropathic pain—in detail, the inferior alveolar nerve transection (IANX) inducing ectopic allodynia behavior in the whisker pad of mice [102]. IANX also promoted the production of peripheral sensitization-related molecules, such as BDNF. Notably, neither high-dose nor low-dose NMDA upregulated BDNF in the TG. Again, in TG, NMDA receptor subunit subtype 2A (NR2A) knockout produced a significant blockade of the IANX-induced changes in BDNF and other molecules. On the contrary, NR2B knockout had no effect on BDNF. Above all, for the peripheral sensitization in the TG, NR2A and NR2B played a similar role in regulating the changes of interleukin 1β, TNF-α, and inward potassium channel; whereas it showed the opposite effect in mediating the production of BDNF and CCL2 (monocyte chemoattractant protein-1, MCP-1).

More recently, the involvement of GABAergic, glutamatergic, and opioidergic systems and BDNF levels was evaluated in the trigeminal neuropathic pain process in rat facial mechanical hyperalgesia induced by chronic constriction injury of the infraorbital nerve (CCI-ION) [103]. BDNF levels were evaluated in the cerebral cortex, brainstem, TG, infraorbital branch of the trigeminal nerve, and serum. Cerebral cortex and brainstem BDNF levels increased in the CCI-ION and sham-surgery groups. Only the CCI-ION group presented high levels of BDNF in the TG. These results suggested the involvement of GABAergic, glutamatergic, and opioidergic systems and peripheral BDNF in the trigeminal neuropathic pain process [103].

### 2.3. Neurotrophins in Brainstem Trigeminal Nucleus

The data reported was obtained from 10 original articles and 0 clinical trials.

#### 2.3.1. Nociceptive Orofacial Pain Modulation

In this section, 4 original research articles on animal models and 0 clinical trials were included and analyzed. 

The effect of trigeminal pain on cyclooxygenase-2 (COX-2) and BDNF expression was evaluated in the Sp5C in a rat model of nociceptive orofacial pain induced by subcutaneous injection of capsaicin in the upper lip [104]. In addition, orexin receptor 1 (OX1R) agonist (orexin-A) and antagonist (SB-334867-A) were administrated in the Sp5C to investigate the possible roles of OX1R on changes in COX-2 and BDNF levels following pain induction. The results presented an increase in COX-2 and a decrease in BDNF immunopositivity in the Sp5C of capsaicin and capsaicin-pretreated with SB-334867-A rat groups. However, the effect of capsaicin on COX-2 and BDNF expressions was reversed by Sp5C microinjection of orexin-A [104]. Therefore, orexin-A may reduce trigeminal pain as well as inflammatory and trophic factors, including COX-2 and BDNF, in the Sp5C [104]. An innovative non-pharmacological approach of transcranial direct-current stimulation was studied to evaluate its effect in rats subjected to a temporomandibular inflammatory pain model [105]. The treatment reduced mechanical and thermal hyperalgesia and also the BDNF and NGF expression in the brainstem, correlating these molecules to nociceptive transmission [105].

Another work, which was previously described in the TG section, demonstrated that in the orofacial pain induced by experimental tooth movement in rats, the intra-TG injection of NGF induced the upregulation of CGRP in both TG and Sp5C [94]. Moreover, in SH-SY5Y neurons, NGF significantly increased synaptic vesicle release and CGRP expression [94]. 

Finally, an alleviation of trigeminal nociception using p75NTR-targeted lentiviral interference therapy was performed to achieve targeted depletion of TRPV1 in TG sensory neurons in rats with positive and interesting results [92]. This research was previously described in the TG paragraph, underling that the Sp5 of the brainstem was also investigated revealing much higher expression of TrkA in Sp5 as compared to p75NTR with the opposite trend in TG [92].

#### 2.3.2. Neuropathic Orofacial Pain and Migraine Pain Modulation

In this section, 6 original research articles and 0 clinical trials were included and analyzed.

Very recently, an in vivo study using a rat model of chronic migraine established by repeated injections of nitroglycerin [106] suggested that pituitary adenylate cyclase-activating peptide (PACAP) induces migraine-like attacks and may potentially be a new target for migraine treatment, but the therapeutic results of targeting PACAP and its receptors are not uniform. The study showed that PACAP and PACAP type 1 receptor (PAC1R) expression were significantly raised in the Sp5C after repeated nitroglycerin injections. Therefore, the synthesis of BDNF rose when synaptic activity increased, indeed BDNF expression levels significantly increased after nitroglycerin injection, and the PAC1R antagonist, PACAP6-38, reversed the effects [106]. These results indicated that PACAP6-38 was capable of improving synaptic transmission and neuronal activation involving BDNF, the extracellular signal-regulated kinase, and the cAMP response element-binding protein (BDNF-ERK-CREB) pathway in chronic migraine rats [106]. The BDNF-ERK-CREB pathway is fundamental for neuronal survival, synapses, and synaptic plasticity and it is also involved in cognitive impairment [107,108].

The contribution of the purinergic P2X4 receptor was investigated together with the related signaling pathways in an inflammatory soup-induced trigeminal allodynia in a rat model, which closely mimics chronic migraine status [109]. The upregulation of the purinergic P2X4 receptor was observed in Sp5C microglial cells. Blockage of the purinergic P2X4 receptor produced an anti-nociceptive effect, with inhibition of BDNF expression among other tested molecules. In addition, double immunostaining indicated that BDNF was mainly expressed in microglial cells.

Moreover, the induced trigeminal neuralgia using CCI-ION was investigated in a rat model [110]. A single administration of low-dose naltrexone, a competitive opioid receptor antagonist, partially reversed facial mechanical allodynia induced by CCI-ION in rats and, after 10 days of treatment, this reversion was complete. Interestingly, low-dose naltrexone was able to variably modulate the levels of BDNF with a decrease in the pain group at the spinal cord level, while in the brainstem, it was unaffected [110]. 

Furthermore, the role of BDNF was explored in a rat model of status epilepticus, induced by intraperitoneal injection of lithium chloride-pilocarpine, and migraine, induced by repeated dural injections of inflammatory soup [111]. The lithium chloride-pilocarpine combination has been extensively used to obtain an animal model of status epilepticus. The lithium-chloride-pilocarpine-treated animals are characterized by staring, head bobbing, blinking, and wet-dog shakes; seizures subsequently appeared 30 min after treatment, each lasting approximately 30–45 s and recurring every 2–5 min [112]. The obtained data indicated that epilepsy favored migraine axis-mediated microglial activation in the cortex/thalamus/Sp5C accompanied by the BDNF release. This information could be used to develop potential therapeutic strategies for preventing and treating migraine in patients with epilepsy [111].

In addition, the regulatory effect of NMDA receptors in mediating peripheral and central sensitization in a mice model of orofacial neuropathic pain involving Sp5C has been described previously in the TG paragraph [102]. 

Finally, the involvement of GABAergic, glutamatergic, opioidergic systems, and BDNF levels has been evaluated in the rat trigeminal neuropathic pain process induced by CCI-ION [103]. This research was previously described in the TG paragraph. In particular, cerebral cortex and brainstem BDNF levels increased in the CCI-ION and sham-surgery groups, conversely only the CCI-ION group presented high levels of BDNF in the TG [103].

In Table 1 the key findings and possible clinical implications are summarized.

## 3. Discussion

This literature screening considered the last ten years and shows a clear trend in the chosen topics to deepen the mechanisms underlying orofacial nociceptive transmission. Regarding peripheral tissues, the topic of the examined articles was prevalently on nociceptive pain both in animal models and in humans. The majority were on NGF, which included clinical trials (Table 2), and one article evaluated both BDNF and NGF in nociceptive pain [82]. Furthermore, in TG, the same trend was observed. Indeed, nociceptive pain was the most considered using animal models or an in vitro approach. Nevertheless, the BDNF topic was preferred with respect to NGF, even if by a little. One considered both BDNF and NT-3 in nociceptive pain [89], and others investigated neurotrophins’ receptors [77,92,98]. Finally, in the brainstem, the trend was different. The topic of the articles was primarily on neuropathic pain in animal models, investigating mostly BDNF. One was about neurotrophins’ receptors [92] and one treated both BDNF and NGF [105]. Some of these articles overlapped because they were interested in more than one anatomical district.

The observations above underline that TG, nociceptive pain, and BDNF represent the most considered target topics over the last 10 years. Regarding TG, it represents a fundamental gating step in nociceptive transmission influencing and modulating both peripheral tissues and brainstem areas via anterograde and retrograde transport of neurotrophins. Moreover, it is necessary to consider the difficulty in isolating the TN relative to TG, which could disincentivize research in this anatomical area. Moreover, BDNF is considered a key molecule in nociceptive peripheral sensitization [102], and it is normally expressed in TG neurons [113]. BDNF released from small-diameter TG neurons upregulates TrkB through a paracrine/autocrine mechanism, increasing spontaneous activity and triggering the release of chemical neuromodulators (e.g., CGRP) that act on the neighboring neurons or satellite glial cells via a similar mechanism [48]. Finally, the study of nociceptive pain could address the difficulty to develop adequate animal models because orofacial neuropathic pain represents a major public health concern with a gradually increasing prevalence in the population [114] and so should be the main target in orofacial pain research.

Regarding the studies in humans, three clinical trials were performed on nociceptive pain in peripheral tissues [80,81,85] and one on neuropathic pain performing genotyping [97]. No human studies were conducted on the brainstem. In addition, the only neurotrophin considered was NGF or its receptor. In this aspect, the peripheral tissue is very accessible with limited disadvantages for patients. Moreover, NGF in nasal biopsy specimens with idiopathic rhinitis (IR) was not altered [80]. Repeated intramuscular injections of NGF in healthy subjects caused an increase in mechanical sensitivity for the masseter muscle but not for the temporalis muscle; nevertheless, both referred pain frequency and number of headache days were not increased following NGF injections [81]. In addition, NGF injection in the masseter muscle of bruxers and in controls showed a significant impact on the corticomotor excitability only in the control group. In turn, NGF sensitization had some impact on force-control mechanisms and masseter-muscle performance and positively influenced the occurrence of significantly higher jaw-pain intensity and limitation, in both bruxers and controls, which most likely aim to protect the musculoskeletal orofacial structures [85]. Finally, another study showed no association between TrkA gene polymorphisms and trigeminal neuralgia in humans [82]. Taken together the results in clinical trials indicate a not determinant role of NGF but suggest a protective role in nociception [85]. 

In addition, two research studies were conducted on human samples [82,83]. The first one [82] reported a decrease in NGF and BDNF in the saliva and only an increase in plasma BDNF in patients with TMD-myalgia. The second one [83] studied and found a sex-related difference in masseter muscles NMDA-receptors expression in nerve fibers. Females showed a higher expression relative to males. The results of these two studies in humans better support the role of NGF and BDNF in nociception compared to the clinical trials.

Regarding the protective role of NGF suggested by Boscato et al. [85] in a clinical trial, the same concept has been proposed in another work using FD-PRP in the axotomized infraorbital nerve in rats [88]. They observed an increase in NGF after 21 days from surgery, indicating the regenerative role of this neurotrophin. Moreover, in the same animal model, it has been suggested a balanced role in the neuroprotection of neurotrophins and their receptors in TG after neurostimulation [98].

Another important aspect proposed during the last ten years of studies is the sex difference in the pathological mechanism of pain, which has a defined biological basis described in several research works, e.g., for neuropathic pain [115,116]. They found that the involvement of spinal microglial BDNF in the induction of mechanical allodynia after nerve injury was male-specific [115]. In addition, 17-β-estradiol’s influence on TMD disorders has been explored considering that women with reproductive capability are particularly affected by this kind of pain [79]. The results showed that 17-β-estradiol upregulated the expressions of TRPV1 and NGF in a dose-dependent manner [79]. In addition, it has been found that NGF increased NR2B-expressing rat trigeminal masseter ganglion neurons in both sexes, but the neuropeptides (CGRP and SP) were increased in NR2B-expressing masseter ganglion neurons only in female rats [78].

The last aspect emerging from this literature review is no consensus among the overall data examined. In particular, some results do not show a determinant involvement of neurotrophins in nociception, as described above in clinical trials. For example, the modulation of BDNF in rat models of trigeminal neuropathic pain was detected in TG [103] but not at the central level [103,110], suggesting a peripheral key role [103]. Moreover, a link between NGF and human idiopathic rhinitis was not found [80]. Furthermore, a correlation between skin peripheral NGF increase, and hyperalgesia was not detected; nevertheless, NGF could contribute to altered neurochemistry [87]. On the contrary, it has been demonstrated that intraganglionic administration of NGF upregulated and downregulated CGRP in TG and Sp5C, respectively [93]. Based on this, it has been hypothesized that following NGF-mediated orofacial pain induced by experimental tooth movement, CGRP synthesized in TG might be transported to Sp5C through vesicle transport [93]. In addition, a decrease in BDNF expression was reported in Sp5C after experimental induction of nociceptive orofacial pain by subcutaneous injection of capsaicin in the upper lip in a rat model [104].

Regarding migraine, Zhou et al. [111] confirmed the important role of BDNF in sustaining this pain state.

Finally, no works on the specific topic of neurotrophin’s role in orofacial pain during the COVID-19 pandemic have been published.

## 4. Conclusions

The literature screening summarized in the present review highlighted the following five main points: (1) TG, nociceptive pain, and BDNF represent the most considered target topics investigated during the last 10 years; (2) few clinical trials were performed on nociceptive or neuropathic pain and none regarding the brainstem. In addition, the only neurotrophin evaluated was NGF or its receptor; (3) the orofacial pain pathological pathway presents a potential sex-related difference; (4) the precise modulation of neurotrophins signaling could switch their role from “promoter” to “inhibitor/blocker” of pain throughout their regenerative potential especially in neuropathic pain; and (5) there is no complete consensus among the overall data reported in the literature. In conclusion, neurotrophins are an interesting target for orofacial pain modulation but more deepened studies are necessary to clarify their role for future application in clinical practice.

## 5. Future Directions

Future research directions may also be highlighted to clarify the role of neurotrophins, especially since they seem to act differently according to the kind of pain and the anatomical site of action, probably also following a specific timeline. Moreover, the overall results could address new and targeted therapeutic strategies developed for clinicians, for example, the use of innovative drugs or different management of the current treatment could have a multidisciplinary approach.

## Figures and Tables

**Figure 1 ijms-24-12438-f001:**
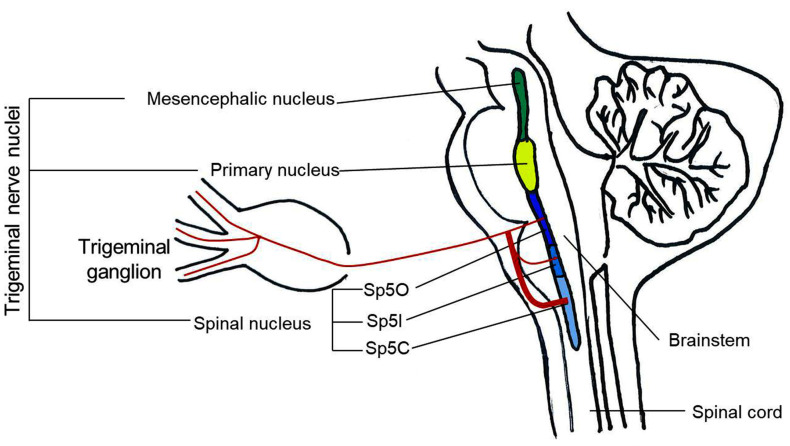
Schematic illustration of the principal anatomical structures involved in orofacial nociception. Sp5O: spinal trigeminal nucleus oralis; Sp5I: spinal trigeminal nucleus interpolaris; Sp5C: spinal trigeminal nucleus caudalis.

**Figure 2 ijms-24-12438-f002:**
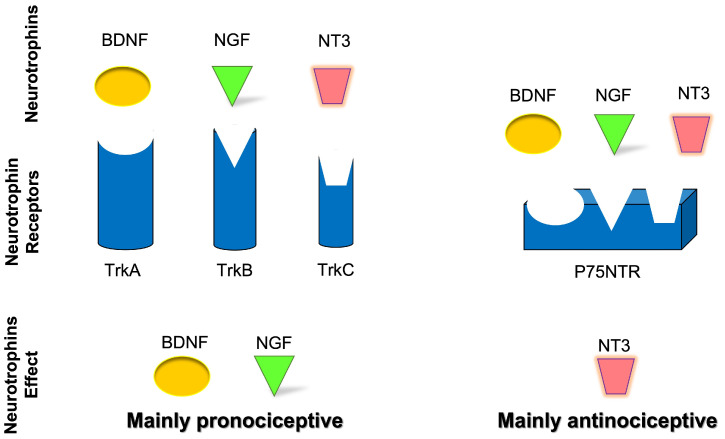
Schematic representation of neurotrophins’ role and effect in nociceptive transmission.

**Figure 3 ijms-24-12438-f003:**
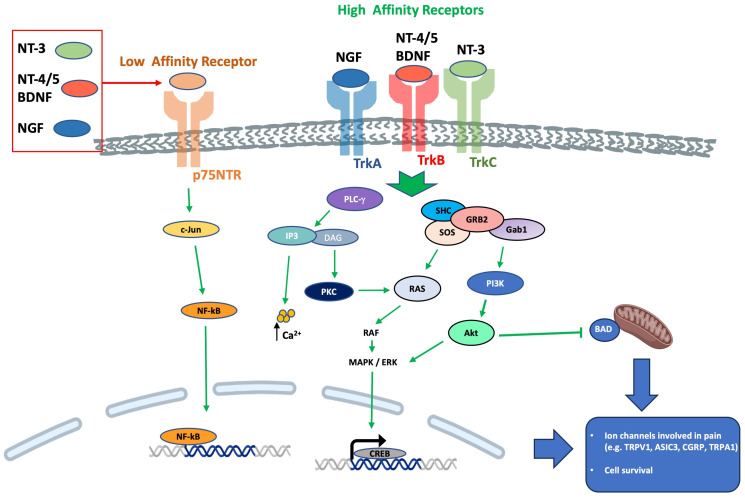
Neurotrophin receptors downstream signaling. Abbreviations: Akt, protein kinase B; BAD, BCL-2 associated death promoter; Ca^2+^, calcium ions; c-Jun, transcription factor Jun; CREB, cAMP response element-binding protein; DAG, diacylglycerol; ERK, extracellular-signal-regulated kinase; Gab1, GRB2 associated binding protein 1; GRB2, growth factor receptor-bound protein 2; IP3, inositol tris-phosphate; MAPK, mitogen-activated protein kinase; NF-kB, nuclear factor kappa-light-chain enhancer of activated B cells; p75NTR, neurotrophin receptor at 75 kDa; PI3K, phosphatidylinositol 3-kinase; PLC-γ, phospholipase C-gamma; RAF, rapidly accelerated fibrosarcoma protein; RAS, small guanosine 5′-triphosphate (GTP)-binding protein; SHC, shc transforming protein; SOS, sons of sevenless; Trk, tyrosine kinase receptor.

**Figure 4 ijms-24-12438-f004:**
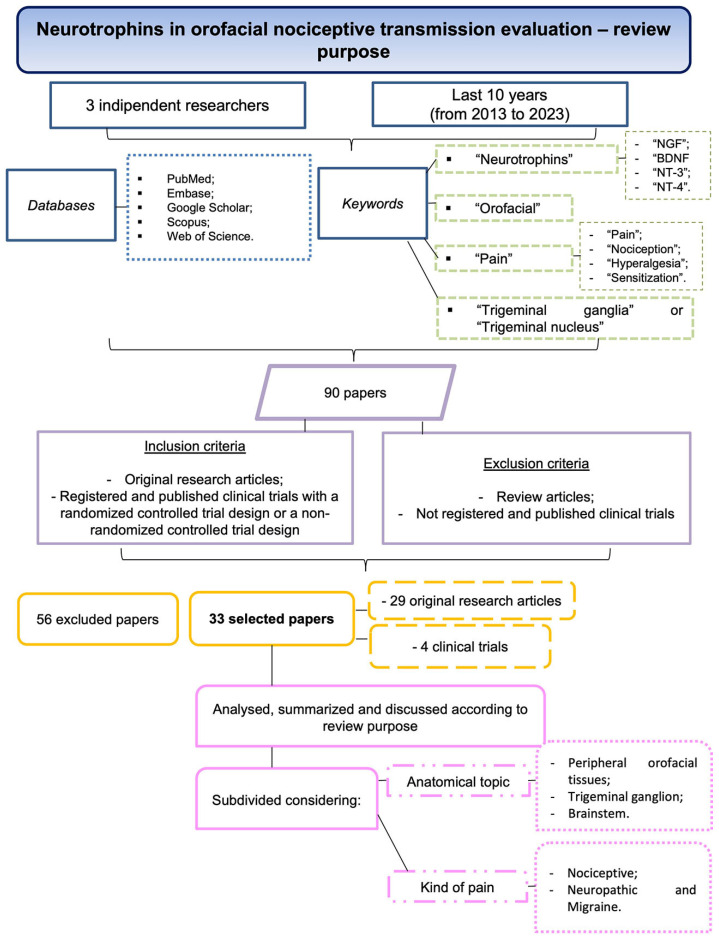
Flowchart summarizing the screening process of the articles.

**Figure 5 ijms-24-12438-f005:**
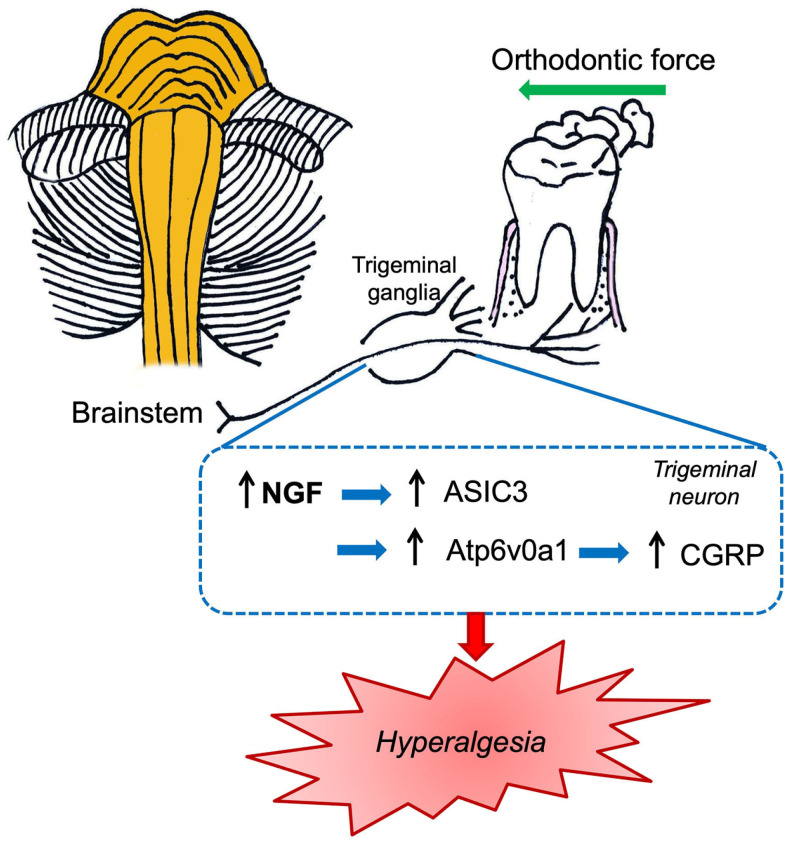
Schematic illustration of the nociceptive mechanisms induced by orthodontic force which summarized the data reported by Gao et al. [84] and Tao et al. [94].

**Table 1 ijms-24-12438-t001:** Summary of key findings and possible clinical implications.

Reference	Key Findings	Possible Clinical Implications
Alhilou et al., 2021 [83]	NGF injection into the human masseter muscle displays a greater magnitude of NGF-induced mechanical sensitization in women compared to men. This sensitization is associated with nerve fibers expression of NMDA-receptors.	Women could be at greater potential risk of developing NGF muscle pain than men.
Benedet et al., 2017 [17]	DREAM regulates trigeminal nociception in TG, at least in part, through the control of BDNF expression levels.	Evidence of BDNF and DREAM role in trigeminal nociception and individuation of new potential molecular targets.
Boscato et al., 2022 [85]	Bruxers do not significantly change the central modulation of motor pathways after NGF injection in masseter muscle while it is altered in control subjects.	NGF-induce sensitization could have therapeutic implications for the potential to “detrain” and manage bruxism
Canner et al., 2014 [89]	First-time description of a relationship between NT-3 and TRPA1 regulation.	Individuation of NT-3/TRPA1 signaling as a potential target in nociception.
Chung et al., 2016 [90]	Masseter inflammation alters the expression of multiple nociceptor genes, among which *Bdnf*, is involved in craniofacial hyperalgesia.	Identification of multiple novel potential pharmacological targets in persistent craniofacial muscle pain.
Costa et al., 2019 [97]	No association is found between TrkA gene polymorphisms and trigeminal neuralgia in humans.	The data do not exclude the possibility that other genotypes affecting the expression of TrkA are associated with the disease.
de Oliveira et al., 2020 [110]	Low-dose naltrexone exerts an analgesic effect in a trigeminal neuralgia rat model involving BDNF modulation in the spinal cord but not in the brainstem.	Administration of low-dose naltrexone may be an option for treating trigeminal neuralgia.
Evans et al., 2014 [87]	Temporal mismatch among the behavioral signs of rat neuropathic pain, skin NGF, and phenotypic changes in cutaneous nerve fibers expressing TrkA.	Skin NGF and nerve fiber TrkA increase are not sufficient to cause hyperalgesia. These results should be considered in future therapeutic approaches.
Exposto et al., 2018 [81]	- NGF repeated injections into the human masseter muscle but not in the temporalis muscle cause mechanical sensitization compared to contralateral saline injections. - NGF injections do not cause an increase in pain frequency and in the number of headache days.	Healthy individuals’ referred pain following palpation may be an epiphenomenon of the muscle in response to NGF noxious input. NGF seems not to be involved in the nociceptive pathway.
Finamor et al., 2023 [103]	BDNF is increased in rat TG in a trigeminal neuralgia model. In the brainstem, BDNF increased also in sham-operated rats.	Inhibition of local BDNF could be a target of future therapies for trigeminal neuralgia.
Gao et al., 2021 [84]	Tooth mechanical hyperalgesia is alleviated by NGF-neutralizing antibody injection in TG.	NGF-based gene therapy is a viable method for alleviating tooth mechanical hyperalgesia.
Grayson et al., 2022 [86]	- BDNF is expressed and released in the tongue tumor mouse model. - BDNF or TrkB inhibition reduces mechanical hypersensitivity and pain-related behaviors in the xenograft model.	BDNF contributes to pain-like behaviors at the site of tumor growth. Inhibitors for BDNF/TrkB could have significant potential in pain treatments allowing for considerable improvement in head and neck cancer patient quality of life.
Guo et al., 2021 [96]	- NGF positively regulates CCL19 and other inflammatory pathways. - CCL19 in TG plays an important role in tooth-movement-induced orofacial pain in a rat model.	Tooth-movement orofacial pain may be modulated by NGF through CCL19.
Jasim et al., 2020 [82]	- Patients with a diagnosis of TMD-myalgia have significantly higher levels of salivary and plasma glutamate as well as plasma BDNF and NGF compared to healthy pain-free individuals. - There is no correlation between subjective pain level and plasma or salivary glutamate, NGF, and BDNF levels.	Potential use of salivary BDNF and NGF as indicative biomarkers for TMD-myalgia.
Kooshki et al., 2018 [104]	- Capsaicin-induced trigeminal pain in a rat model affects COX-2 and BDNF expression in Sp5C. - OX1R agonist (orexin-A) can attenuate capsaicin-mediated effects.	Orexin-A may be a potential treatment for trigeminal pain.
Kovačič et al., 2013 [77]	- Most pulpal and gingival neurons express the TrkA receptor. - Sensory neurons of dental pulp are larger in comparison to the TG neurons projecting to gingivomucosa. Both tissues are richly innervated by NGF-dependent nociceptive neurons. - The great majority of TG neurons innervating the dental pulp are NGF-dependent A-fibre nociceptors.	Different sensitivity of gingivomucosa and dental pulp during normal and pathological conditions suggests its potential clinical applications in tissue-specific modulation of nociception.
Liu et al., 2018 [109]	- IS induces Sp5C microglia activation and purinergic P2X4 receptor upregulation in a rat model of chronic migraine. - The following BDNF release and consequential TrkB-activation affect inflammation-induced trigeminal allodynia.	Microglia activation, modulation of purinergic P2X4 receptor activation, and BDNF-TrkB signaling are potential therapeutic targets for treating trigeminal allodynia.
Liu et al., 2020 [101]	Palmatine administration increases the mechanical pain threshold in a rat model of trigeminal neuralgia by reducing the expressions of BDNF and TrkB and by inhibiting the ERK1/2 pathway in TG.	Palmatine and its analgesic mechanism could act as a potential pharmacotherapy in the treatment of trigeminal neuralgia and other chronic pain conditions.
Luo et al., 2020 [100]	- BDNF and GDNF are involved in the nociceptive excitability of TG neurons in a rat model of trigeminal neuralgia. - BDNF and GDNF affect the transmission of peripheral orofacial nociceptive stimuli from afferent fibers through the trigeminal root entry zone to the brainstem and higher-level brain regions.	BDNF and GDNF signaling could be pharmacological targets in trigeminal neuralgia.
O’Leary et al., 2018 [92]	p75NTR targeted lentiviral interference therapy is proposed to alleviate trigeminal nociception to achieve targeted depletion of TRPV1 in rat TG sensory neurons with novel and positive results monitoring TG and Sp5C.	This study introduced a therapeutic targeting strategy that provided a means to post-transcriptionally downregulate TRPV1 in TG nociceptors of adult rats in vivo.
Rahmi et al., 2022 [88]	- FD-PRP injection in a rat model of infraorbital axonotmesis injury increases significantly NGF and S100B levels at the infraorbital nerve level. - FD-PRP shortens the process of acute inflammation so the process of nerve regeneration can initiate quickly.	Neuroregeneration is a process needed for the treatment of neuropathic pain. FD-PRP injection is effective in inducing neuroregeneration by increasing NGF and S100B expression.
Scarabelot et al., 2019 [105]	Transcranial direct-current stimulation reduces mechanical and thermal hyperalgesia and also the BDNF and NGF increase in the brainstem of a rat model of TMJ pain.	Transcranial direct-current stimulation may be a non-pharmacological and non-invasive therapeutic tool against orofacial pain.
Takeda et al., 2013 [48]	BDNF enhances the excitability of the small-diameter TG neurons projecting onto the Sp5I/Sp5C following a rat model of masseter muscle inflammation.	TG BDNF-TrkB signaling could be a therapeutic target for the treatment of trigeminal inflammatory hyperalgesia.
Tao et al., 2022 [94]	In an experimental model of orofacial pain induced by tooth movement in rats, NGF regulates CGRP expression both in TG and Sp5C.	NGF and CGRP are involved in the transmission of nociceptive information in orofacial pain, so they could be the targets for future therapies.
Van Gerven et al., 2017 [80]	- TRPV1 and TRPA1 nociceptors are fundamental in the pathophysiology of human idiopathic rhinitis. - Capsaicin treatment reduces the symptoms, but NGF is not altered in nasal biopsy specimens both in idiopathic rhinitis patients and in healthy control subjects.	NGF seems not involved in idiopathic rhinitis and so it couldn’t be considered a potential therapeutic target.
Virtuoso et al., 2019 [98]	Following nerve axotomy in a rat model and electrical stimulation of the infraorbital nerve, RET- and Trk-expression patterns indicate that sensory TG neurons express NGF, BDNF/NT-4, GDNF, and NT-3 receptors at levels similar to those found in physiological conditions, although they have presumably switched to regeneration-repair state due to the injury.	Neurostimulation protocols, either for therapeutic applications in neuropathic pain or for the development of nerve-machine sensory neuroprostheses, should be designed considering the sensory modality of target-ganglion neurons and the specific alterations they will elicit on each fiber/neuron type.
Wang et al., 2019 [95]	BDNF/TrkB enhances the T-type channel through the PI3K-p38-PKA signaling cascade resulting in TG pain hypersensitivity in rats.	BDNF-TrkB pathway in TG neurons may be a tool for developing pain therapeutics in clinical applications.
Wang and Chung, 2020 [93]	- An orthodontic force producing mechanical irritation and localized inflammation in the periodontium induces upregulation of pain-related genes, including *Bdnf*, in mice TG. - Transcriptomic changes resembling nerve injury occur in the TG upon application of orthodontic force. - The injury is driven by nociceptive inputs through TRPV1.	The results lead to the identification of targets for better management of pain and sensory disturbances during orthodontic treatment.
Wong et al., 2014 [78]	- NGF induces long-lasting mechanical sensitization in masseter muscle in rats. It is major in female than male rats. - DL-2-amino-5-phosphonovaleric acid reversed NGF-induced mechanical sensitization only in male rats. - NGF-induced sensitization of masseter nociceptors is mediated, in part, by enhanced peripheral NMDA receptor expression in rats of both sexes.	- Chronic pain conditions amenable to treatment with anti-NGF antibodies may also respond to treatments that target the peripheral NMDA receptors. - The peripheral NMDA receptors may be considered biomarkers for myofascial temporomandibular disorders. - Potential sex-related differences in NGF-induced mechanical sensitization.
Wu et al., 2015 [79]	- Estradiol upregulates in a dose-dependent manner the expressions of synovial TRPV1 and NGF. - Activation of TRPV1 enhances cyclooxygenase-2 transcription. - TRPV1 antagonist, capsazepine, attenuates allodynia of the inflamed TMJ.	The study results could potentially help clinicians understand the sexual dimorphism of TMD pain.
Zhang et al., 2023 [106]	- PACAP, PAC1R, and BDNF expression are significantly raised in the Sp5C after repeated nitroglycerin injections as a chronic migraine rat model. - A PAC1R antagonist, PACAP6-38, improves pain thresholds, restores aberrant synaptic ultrastructure, and regulates the ERK/KREB/BDNF signaling.	The findings may suggest PACAP/PAC1R inhibition as a potential therapeutic target for migraine.
Zhang et al., 2022 [102]	Differential roles of NR2A and NR2B in mediating peripheral sensitization in the TG and central sensitization in the Sp5C and contributing to orofacial neuropathic pain in a mice model also influencing BDNF release.	The results may be a fundamental basis for advancing knowledge of the neural mechanisms’ reaction to nerve injury with future translational research in clinical studies.
Zhou et al., 2022 [111]	The data indicate that epilepsy favors migraine axis-mediated microglial activation in the cortex/thalamus/Sp5C and is accompanied by the BDNF release.	This information could be used to develop potential therapeutic strategies for preventing and treating migraine in patients with epilepsy.

**Table 2 ijms-24-12438-t002:** Summary table of the neurotrophins’ involvement in orofacial pain with respect to anatomical structures (peripheral tissues, trigeminal ganglion), and trigeminal nucleus—Sp5C), and the type of pain with the related references of the last ten years (2013–2023).

Anatomical Structures	Type of Pain	Neurotrophins	References
Peripheral tissues	Nociceptive pain	BDNF	[77,78,79,80,81,82,83,84,85,86]
		NGF	
	Neuropathic pain and migraine	BDNF	[87,88]
		NGF	
Trigeminal ganglion	Nociceptive pain	BDNF	[17,48,77,78,84,89,90,92,93,94,95]
		NGF	
	Neuropathic pain and migraine	BDNF	[81,96,97,98,100,101,103]
		NGF	
Trigeminal nucleus (Sp5C)	Nociceptive pain	BDNF	[92,94,104,105]
		NGF	
	Neuropathic pain and migraine	BDNF	[102,103,106,109,110,111]
		NGF	

## Data Availability

No new data were created or analyzed in this study. Data sharing is not applicable to this article.

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
