# Peer review of "Role of Neurotrophins in Orofacial Pain Modulation: A Review of the Latest Discoveries"

_ijms, 2023, doi:10.3390/ijms241512438_

Round 1
Reviewer 1 Report
The topic is very interesting!
However the way of reporting for this review of the last discoveries in role of neurotrophins in orofacial pain modulation is very weak!
Authors have to write professional Introduction and highlight why this review is novel and important and useful. Please rationale the review.
Furthermore, Authors have to precisely describe the methodology of the literature review performing (please add Materials and Methods section). Especially, used databases, inclusion and exclusion criteria for selected literature, and time frame. Please explain how Authors define the last discoveries in Materials and Methods?
Authors can consider to write that pandemic affect the orofacial pain (citing these articles is voluntary) and it could be neurological and psychological orofacial pain modulation eg.
Saki M, Shadmanpour M, Zarif Najafi H. Are individuals with orofacial pain more prone to psychological distress during the COVID-19 pandemic? Dent Med Probl. 2021;58(1):17–25. doi:10.17219/dmp/131683
Emodi-Perlman A, Eli I, Smardz J, Uziel N, Wieckiewicz G, Gilon E, Grychowska N, Wieckiewicz M. Temporomandibular Disorders and Bruxism Outbreak as a Possible Factor of Orofacial Pain Worsening during the COVID-19 Pandemic-Concomitant Research in Two Countries. J Clin Med. 2020 Oct 12;9(10):3250. doi: 10.3390/jcm9103250.
Authors have to prepare the comprehensive table which presents the key findings from the literature review and highlight the most important issue for clinical practice.
The language has to be revise by native speaker after all corrections.
Reviewer 2 Report
Bonomini, Favero and collaborators review a role of neurotrophins in modulating orofacial pain transmission. Fig. 1 shows a schematic diagram of neurotrophins, their receptors and their effects on pain transmission. Fig. 2 presents an involvement of NGF in hyperalgesia produced by orthodontic force. Table 1 summarizes references examining an involvement of NGF and BDNF in nociceptive, neuropathic pain and migraine at peripheral, trigeminal ganglion and trigeminal nucleus levels. There are several points that should be addressed and may serve to amend this manuscript, as follows:
Major points:
1. A list of the abbreviations used in this review article would be helpful to the reader. Please amend this point.
2. This review article should be written in such a way that the reader can understand what is written without referring to the references cited, as much as possible. Some of the improvements for this are mentioned in the specific comments below.
3. Table 1: it will not be necessary to give “number of published articles”, because this number is uncommon and the authors may have overlooked relevant papers. Please amend this point.
4. First paragraph on page 12: as mentioned above, it seems totally meaningless to give the number of papers examined by the authors. Please amend this point.
Specific points:
1. Line 42: please use either “brainstem” (this line) and “brain stem” (Fig. 2) throughout this manuscript.
2. Line 73: not “SpO and SpI” but “Sp5O and Sp5I”? Please check this point.
3. Lines 107 and 108: please correct the definition of TRPV1.
4. Lines 108 and 109: not “SP and “CGRP” but “SP and CGRP receptors”? Please check this point.
5. Line 112: “KCC2” should be shortly explained.
6. Line 157: not “2.1.1..” but “2.1.1.”.
7. Line 166: please use either “fibre” of “fiber” (line 131) throughout the text.
8. Line 171: not “ug and ul” but “μg and μl”? Please check this point.
9. Line 176: what is “10 ll”? Please make this point clear.
10. Line 181: not “17-b-estradiol” but “17-β-estradiol”.
11. Line 278: not “S100 B” but “S100B” (see line 280). Please explain S100B shortly.
12. Line 284: not “2.2.1..” but “2.2.1.”.
13. Lines 293 and 302: not “trkB” but “TrkB”.
14. Line 315: are “sensations” “membrane proteins”? Please check English.
15. Line 319: is “And is ..” OK? Please check English.
16. Line 331: it will be better to explain shortly DREAM protein.
17. Line 378: not “are” but “is”.
18. Line 402: not “neuro regenerative” but “neuroregenerative”.
19. Line 406: please explain shortly about RET.
20. Line 437: not “2.3.1..” but “2.3.1.”.
21. Lines 449 and 450: it is not clear what “the modulation of pain effects on COX-2 and BDNF expressions” means. Please amend this point.
22. Line 459: it is not clear what “NGF modulated synaptic vesicle” means. Please amend this point.
23. Lines 461 and 462: it is not clear what “chronic migraine induced in a rat model” means. Please amend this point.
24. Line 469: there is no explanation about PACAP6-38. Please amend this point.
25. Line 471: from the results stated here, it is unknown whether ERK/CREB is involved. Please amend this point.
26. Line 487: naltrexone is generally known to be an opioid-receptor antagonist. If this drug is used as an anticonvulsant in the cited paper, a comment should be stated here.
27. Lines 493 and 494: please state shortly the actions of LiCl and pilocarpine.
28. Lines 492-494: is English here OK? Please check English.
29. Line 598: not “with” but “and”? Please check English.
30. There seem to be more mistakes than stated above. Please check the manuscript very carefully.
The quality of English Language is not so good. The authors should amend English of this manuscript.
Reviewer 3 Report
The first part of the review requires additional information and illustrations. The second part 2. The discoveries on neurotrophins of the last 10 years (2013-2023) is beautifully written. Huge work has been done. The discussion of the results is comprehensive and impressive. 1) 1.2. Anatomical structures involved in orofacial nociception. An illustration is needed. 2) It must be clearly stated which brain cells release neurotrophins. 3) A scheme of intracellular signaling of neurotrophins is needed. Relationship with changes in the expression of protective genes, Ca2+ signaling, regulation of mitochondrial activity, etc.
Review written in good language
Reviewer 4 Report
thank you for intersting paper
add review chart flow
highlight the aim of study
conclusions and discussions should be written separately
in conclusion add top 5 most important issues in this review
how the pain control can be handled with clinicians?
add a chapter on pain differential diagnosis - especially trigeminal neuralgia
Round 2
Reviewer 1 Report
The manuscript has been revised correctly. I don't have further comments.
I don't have remarks.
Reviewer 2 Report
This revised manuscript has been amended in response to my comments. There are only minor points that should be considered, as follows:
1. Line 154: not “promote” but “promotes”? Please check English.
2. Lines 187 and 188: there are two definitions of MAPK. Please amend this point.
3. Lines 188: not “kinases” but “kinase”? Please check this point.
4. Figure 3 and its legend: please use either “RAS” (figure) or “Ras” (legend). Please check if the figure and legend correspond.
5. Line 219: is necessary “.” following “such as”? Please check English.
6. Line 233: “4” in “IB4” should be subscript.
7. Line 248: not “Complete” but “complete”? Please check this point.
8. Lines 251, 254, 255, 256, 663 and 665: not “estradiol” but “17-β-estradiol”? Please check this point.
9. Lines 278-280: this is a phrase, but not a sentence. Not “concentration” but “concentrations”? Please check these points.
10. Line 334, Evans et al., 2014 [87] on page 15, Luo et al., 2020 [100] on page 16 and Virtuoso et al., 2019 [98] on page 17: not “fiber” but “fibre” (see lines 102 and 103).
11. Lines 354 and 355: not “NMDA receptor subtype 2B” but “NMDA receptor subunit subtype 2B”? Please check this point.
12. Line 375: not “were” but “was”? Please check English.
13. Line 377: not “TRPA1” but “TRPA1s”?
14. Line 414: not “were” but “was”? Please check English.
15. Line 417: please use either “L” or “l” (see line 241) throughout this manuscript.
16. Line 481: not “were” but “was”? Please check English.
17. Line 492: not “NMDA receptor subtype 2A” but “NMDA receptor subunit subtype 2A”? Please check this point.
18. Lines 557-559: this is a phrase, but not a sentence. Please check this point.
19. Lines 570 and 571: not “lithium chloride pilocarpine” but “lithium chloride and pilocarpine”? Please check this point.
20. Costa et al., 2019 [97] on page 15: not “.. is associated ..” but “.. are associated ..”.
21. Grayson et al., 2022 [86] on page 15: is “.. in of ..” OK? Please check English.
22. Kovačič et al., 2013 [77] on page 16: is “The great majority of all TG neurons” OK? Please check English.
23. Rahmi et al., 2022 [87] on page 17: not “shorten” but “shortens”? Please check English.
24. Van Gerven et al., 2017 [80] on page 17: is “synthomps” OK? Please check English.
25. Zhang et al., 2023 [106] on page 18: not “restore” but “restored”? Please check English.
26. Line 597: please put “.” following “humans”.
27. Line 654: not “PRP” but “FD-PRP”? Please check this point.
28. Abbreviations: does “BAD” appear in this manuscript? Is “P2X4: purinoceptor” OK? Is “PAC1: platelet-activating factor acetylhydrolase ib subunit alpha” OK? Please check these points. Moreover, please check if the words appearing in this abbreviation list are actually used in the text.
There seem to be more mistakes than pointed out above. Please check this revised manuscript very carefully.
English language is necessary to be checked.
Reviewer 4 Report
thank you, looks very nice
